# Mycotoxins in Tea ((*Camellia sinensis* (L.) Kuntze)): Contamination and Dietary Exposure Profiling in the Chinese Population

**DOI:** 10.3390/toxins14070452

**Published:** 2022-07-01

**Authors:** Haiyan Zhou, Zheng Yan, Aibo Wu, Na Liu

**Affiliations:** SIBS-UGENT-SJTU Joint Laboratory of Mycotoxin Research, CAS Key Laboratory of Nutrition, Metabolism and Food Safety, Shanghai Institute of Nutrition and Health, University of Chinese Academy of Sciences, Shanghai 200030, China; zhouhaiyan2018@sibs.ac.cn (H.Z.); zyan@sibs.ac.cn (Z.Y.); abwu@sibs.ac.cn (A.W.)

**Keywords:** mycotoxins, tea, dietary exposure, risk assessment

## Abstract

Tea is popular worldwide with multiple health benefits. It may be contaminated by the accidental introduction of toxigenic fungi during production and storage. The present study focuses on potential mycotoxin contamination in tea and the probable dietary exposure assessments associated with consumption. The contamination levels for 16 mycotoxins in 352 Chinese tea samples were determined by ultra-performance liquid chromatography–tandem mass spectrometry. Average concentrations of almost all mycotoxins in tea samples were below the established regulations, except for ochratoxin A in the dark tea samples. A risk assessment was performed for the worst-case scenarios by point evaluation and Monte Carlo assessment model using the obtained mycotoxin levels and the available green, oolong, black, and dark tea consumption data from cities in China. Additionally, we discuss dietary risk through tea consumption as beverages and dietary supplements. In conclusion, there is no dietary risk of exposure to mycotoxins through tea consumption in the Chinese population.

## 1. Introduction

Tea (*Camellia sinensis* L.) has long been consumed, primarily as an aromatic beverage, in many countries worldwide, with the properties of multiple health benefits and good taste [1,2,3]. The application and consumption of commercial dried tea in many fields is also gradually expanding due to the continuous innovation of tea comprehensive processing technology, forming diversified tea products [4]. Botanical dietary supplements based on traditional raw tea materials have entered our market in different forms [5,6]. The global average consumption of dried tea is approximately 2.5 million metric tons per year [7]. China plays a vital role among the major tea-producing countries. According to the differences in processing technologies and fermentation degrees, Chinese tea can mainly be divided into green tea (undergoing steaming and without fermentation), dark tea (undergoing steaming and postfermentation), oolong tea (semifermentation, not undergoing steaming) and black tea (complete fermentation, not undergoing steaming) [8].

Mycotoxins in tea, as potential chemical pollutants, are of great concern because of the incidental introduction of toxigenic fungi during the long manufacturing and supply of tea from cultivation to final consumption [9,10,11,12]. Filamentous fungi have been found in the postfermentation and storage processes, and some are essential contributors to the quality of dark tea [13,14,15,16]. However, there are concerns that some strains of *Fusarium* spp., *Aspergillus* spp., and *Penicillium* spp. from other environments might be OTA (ochratoxin A), AFs (aflatoxins), ZEN (zearalenone), DON (deoxynivalenol), or CIT (citrinin) producers [17,18,19]. The associated health risks from the potential multiple mycotoxin contamination of raw tea materials used for beverages or dietary supplements have also received increasing research attention [6,20,21]. Many countries have made their own regulations on AFB_1_ and AFs in raw tea materials [22].

In recent years, there has been progress in the quantitative detection of mycotoxins in tea [5,6,12,22]. Many scholars have also focused on risk assessments of AFs and OTA in dark tea [23,24], AFs in black tea [25], and AFs and ZEN in green tea, black tea, and even herbal teas [26]. However, there are limited studies on the risk assessment of multiple mycotoxins involved in tea (*Camelia Sinensis* L.) [27,28,29] compared to herbal teas [30,31,32,33,34,35]. The total risk of exposure to ZEN is also rarely assessed, underestimating the potential toxicity of biotransformation through biochemical components or fungi [36].

Thus, this work aims to investigate and comparatively analyze the contamination levels for 16 mycotoxins from 7 groups in 352 Chinese tea samples (81 green tea, 64 oolong tea, 104 black tea, and 103 dark tea) using UPLC–MS/MS (ultra-performance liquid chromatography–tandem mass spectrometry). In this study, considering the different human preferences and needs regarding the aroma, flavor, taste, and active constituents, risks of dietary exposure to mycotoxins through various types of tea consumption are assessed by point evaluation and the Monte Carlo assessment model, using mycotoxin levels and available tea consumption data obtained from most cities in China. A risk ranking of both tea categories and mycotoxins is also performed. Moreover, the nonneoplastic and neoplastic toxicities of OTA and the carcinogenic risks of individual lifetime AF exposure through drinking tea are also considered. In particular, the risk of green tea powder as a dietary supplement is evaluated preliminarily for Chinese adults.

## 2. Results and Discussion

### 2.1. Occurrence of Mycotoxins

A total of 352 Chinese tea samples were detected and analyzed for 16 mycotoxins in the present study. Descriptive statistics of the concentrations of 16 mycotoxins in different tea categories are summarized in Appendix A. In general, the obtained results demonstrate that the occurrence of mycotoxins was different in various types of tea samples, while DON and OTA had significant differences. This may be influenced by various processing methods or biochemical components. Green tea, without a fermentation process and containing abundant polyphenols, had the lowest levels of mycotoxin contamination, while dark tea had a higher possibility of contamination during long-term processing and storage. We did not exclude the influence of differences in the detailed tea varieties and other complicated factors (brand, source, or date of production). The United States Pharmacopoeia has set the limit for AFB_1_ and AFs in raw medicinal herb materials at 5 and 20 µg·kg^−1^, which are the same as the Canadian and Argentine regulations [6]. Custom union countries (Kazakhstan, Armenia, Kyrgyzstan, Belarus, and Russia) have also established a regulation for the AFB_1_ value in raw tea material at 5 µg·kg^−1^ [37].

In this paper, the content of AFB_1_ in all tea samples did not exceed 5 µg·kg^−1^. The positive detection rates of AFs in dark, black, oolong, and green tea were 6.80%, 1.92%, 0%, and 0%, respectively. The co-occurrence rate of DON and its derivatives (7.39%) in all tea samples was also lower than the result obtained by Reinholds et al. [32]. Their derivatives in all tea samples were far below the MRLs (maximum regulation limits: 750 µg·kg^−1^) for DON in cereals for direct consumption [38]. However, the positive detection rates of DON in dark, oolong, black, and green tea were 7.41%, 9.38%, 2.88%, and 0%, respectively. Bonferroni posttests showed that the concentration of DON in green tea was significantly lower than that in oolong or dark tea (*p* ≤ 0.01). Furthermore, the concentration of DON in oolong tea was also significantly higher than that in black tea (*p* ≤ 0.01). The contents of OTA and CIT in dark tea exceeded 5.0 and 200 µg·kg^−1^. Additionally, the mean concentration of OTA in dark tea (129.76 µg·kg^−1^) was significantly (*p* ≤ 0.01) higher than that in other tea (0.09–1.33 µg·kg^−1^) (Appendix A). The distribution of contamination levels for OTA in different types of tea samples is shown in Figure 1. However, we did not detect high concentrations of ZEN (0.17–2.10 µg·kg^−1^), which is different from Ye et al. [27]. The contents of NEO and T-2 in all tea samples were far below 200 µg·kg^−1^ (Appendix A).

### 2.2. Risk Assessments

#### 2.2.1. Deterministic Estimation

A deterministic assessment of exposure to six groups of noncarcinogenic mycotoxins through four types of tea consumption was completed (Equations (1) and (2)), and is shown in Table 1 and Figure 2. HQ values (lower, middle, and upper bound) for single mycotoxins from various types of tea consumption were far below 1.0, and the HQ values at the upper bound for green tea decreased in the order of ZEN > DON > T-2 > NEO > OTA > CIT. Furthermore, the upper bound HQ values for oolong, black, and dark tea decreased in the order of: ZEN > DON > CIT > OTA > T-2 > NEO; ZEN > DON > OTA > CIT > T-2 > NEO; and OTA > ZEN > DON > CIT > NEO > T-2, respectively. The HI values at the upper bound of the six groups of mycotoxins obtained from the deterministic estimation of green, oolong, black, and dark tea consumption were 1.90 × 10^−2^, 4.41 × 10^−2^, 4.99 × 10^−2^, and 5.46 × 10^−1^, respectively, without significant differences. Moreover, HI values were all <1.0 in the following order: dark tea > black tea > oolong tea > green tea. These values indicated no significant noncarcinogenic health risk from mycotoxins in tea consumption (Figure 2). Statistical analyses suggested significant differences between the upper bound HQ values of OTA in dark tea and green, oolong, and black tea in the deterministic estimation. Meanwhile, concerning the differences in neoplastic and nonneoplastic effects for OTA, the MOE approach for the exposure assessment was performed by Equations (6) and (7). MOE_1_ exceeded 200 in all exposure scenarios, indicating no health risk. Meanwhile, MOE_2_ exceeded 10,000 for almost all exposure scenarios, revealing a low risk apart from dark tea exposure, which indicated a potential health problem when genotoxicity was direct (Table 2). The carcinogenic risk of individual lifetime AF exposure from tea consumption was estimated by Equation (8). The results showed that the carcinogenic risks of individual lifetime AF exposure for four types of tea consumption were far below the acceptable level of 10^−5^ (Table 3).

#### 2.2.2. Probabilistic Estimation

Concerning the variability and sampling uncertainty of the deterministic assessment, a probabilistic evaluation for mycotoxins was necessary to exclude the influence of individual differences, especially for OTA in dark tea. Table 4 shows the exposure distribution statistics of the six groups of mycotoxins for the four types of tea in the mean, 50th, and 95th percentiles performed with Equation (1) (Appendix A). At the same time, the noncarcinogenic risks of different tea types from three bounds of probabilistic estimation (P95) performed by Equation (3) are also shown in Figure 2 and Appendix A.Statistical analyses suggested that there was no significant difference in the MOS values of mycotoxins in green, oolong, black, and dark tea from the probabilistic estimation, which was similar to the above ranking order of deterministic assessments, except for NEO and T-2 in black tea and OTA in dark tea. Interestingly, the upper bound MOS (P95) value of OTA from the probability assessment was significantly lower than the upper bound HQ value from the deterministic assessment, which resulted in a change in tea risk order. In summary, the MOS values at three bounds for the six groups of mycotoxins in the four types of tea were all far below 1.0. Even the HI values calculated by Equation (5) in the four types of tea were still <1.0 (black tea > dark tea > oolong tea > green tea); thus, no significant noncarcinogenic risk was observed from the probabilistic estimation in this study. According to the probabilistic estimation data for OTA, MOE_1_ and MOE_2_ exceeded 200 and 10,000 in all exposure scenarios, also indicating no nonneoplastic and neoplastic concern from OTA exposure through tea consumption (Table 2). The carcinogenic risk of individual lifetime AF exposure through drinking tea was almost below the acceptable level (Table 3). More detailed exposure assessments for AFG_2_ were also performed by a Monte Carlo assessment model using dark tea consumption and body weight data in different age groups from the Kunming, Pu’er, and Ulan Bator of Mongolia questionnaire results (Appendix A). The 95th percentile R values of AFG_2_ in Pu’er for the 30–39 age group (1.13 × 10^−5^) and Ulan Bator and Pu’er for the male group (1.35 × 10^−5^, 1.34 × 10^−5^) and >50 age group (1.52 × 10^−5^, 1.82 × 10^−5^) were close to the acceptable carcinogenic risk level. Interestingly, some of the groups were consistent with the high-exposure groups in previous studies, such as the age group (>50 years old) in Ulan Bator and Pu’er, and the male group in Pu’er. Although contaminant content is the most essential determining factor for mycotoxin exposure in dark tea [27], mycotoxin exposure may be correlated with differences in tea drinking habits and preferences (gender- or age-related).

#### 2.2.3. Dietary Risk Profiling

We started by monitoring raw tea materials and observed no risk of exposure to mycotoxins through drinking tea in the worst-case scenarios, including nonneoplastic or neoplastic risks for OTA and risks of noncarcinogenic and carcinogenic mycotoxins. Information about the change in mycotoxin content after brewing with hot water showed that the transfer rates of these mycotoxins in spiked loose green tea were below 90%, especially AF_S_ (<45%) [39]; the mean transfer rates of AFs in black tea were 30.6% or 53.02% [25,26]. Clearly, there is also no risk of exposure to mycotoxins through drinking tea infused in hot or cold water. In addition to drinking tea, ready-to-eat green tea powder as an alternative strategy for weight loss is becoming increasingly available and is more acceptable to the Chinese population than other forms of dietary supplements (tablets or capsules). The concentrations of mycotoxins in botanical supplements are almost consistent with those in raw materials [6]. Therefore, we also performed a preliminary evaluation of green tea powder using the average body weight of 64.3 kg for adults and the recommended dosage of green tea powder (1000 mg/day). The probable values of EDI (from 1.02 × 10^−6^ to 7.83 × 10^−4^) obtained in any exposure scenario were far below the values of PMTDI established (Appendix A) and the values of EDI for drinking tea in the worst-case scenarios, indicating no risk through the consumption of green tea powder at the recommended dosage for Chinese adults.

## 3. Conclusions

The safety of tea and its consumption has been widely investigated. After the investigation and comparison of mycotoxin contamination levels in major tea categories from Chinese tea samples, differences between green tea and black or dark tea were found due to the existence of high contamination levels in individual varieties from the same category, highlighting the need for early monitoring in the processing process in order to keep the contamination levels as low as possible, especially for fermented tea with complex ingredients or long production chains. Taking into account differences in human preferences and needs, preliminary evaluations found that there is no dietary risk of exposure to mycotoxins through tea consumption as a beverage or dietary supplement. All in all, more up-to-date detailed data on different tea consumption should be encouraged to be recorded in future to further assess the health risks associated with tea consumption. In addition, tea is only one dietary source of possible exposure to mycotoxins, and more comprehensive dietary risk assessments should be carried out in the future based on the contributions of different diets.

## 4. Materials and Methods

### 4.1. Materials and Reagents

ZEN (Art. No. Z 2125), α-zearalenol (α-ZEL, Art. No. Z 0166), β-zearalenol (β-ZEL, Art. No. Z 2000), α-zearalanol (α-ZAL, Art. No. Z 0292), β-zearalanol (β-ZAL, Art. No. Z 0417), DON (Art. No. D 0156), 3-acetyl deoxynivalenol (3-Ac DON, Art. No. A 6166), 15-acetyl deoxynivalenol (15-Ac DON, Art. No. A 1556), OTA (Art. No. O 1877), T-2 toxin (T-2, Art. No. T 4887), and CIT (Art. No. C 1017) were obtained from Sigma-Aldrich (St. Louis, MO, USA). Ammonium acetate, silica gel (SG, Art. No. 236799), and formic acid (≥95%) were also from Sigma-Aldrich. AFs (Part. No. 10000344) and neosolaniol (NEO, Part. No. BRM S 92001) were from Romer Lab Biopure™. MWCNTs-COOH (carboxyl multiwalled carbon nanotubes, Art. No. 190710134732) was acquired from Klamar Reagent, Inc (Shanghai, China). HLB (hydrophilic–lipophilic balance; Art. No. SBEQ-CA3100) was purchased from Anpel Laboratory Technologies. Acetonitrile came from Merck (Darmstadt, Germany) and was HPLC grade.

### 4.2. Sampling and Preparation

A total of 352 tea samples were randomly collected from two representative enterprises: Shanghai Difute International Tea Co., Ltd. (Shanghai, China) and the Yunnan Fengqing Longrun Co., Ltd. (Guiyang, China) Each tea category (green, oolong, black, or dark tea) contained more than 60 tea samples, including at least 10 major tea consumption varieties. Specifically, green tea included Longjing, Maojian, Biluochun, Yunwu, Maofeng and other green tea varieties from different origins; oolong tea included Tieguanyin, Dahongpao and other oolong tea varieties from different origins; black tea included wild black tea, Lapsang souchong, Jinjunmei, Wuyi bohea and other black tea varieties from different origins; dark tea included Puerh tea, Kang brick tea, Fu brick tea, Liubao tea, Tibetan tea and other varieties. The simultaneous determination of mycotoxins was performed using the UPLC–MS/MS method reported by our group [40]. Briefly, 1.0 g sample was ultrasound-assisted extracted using 10 mL acetonitrile with 1% formic acid and 1 mL of the co-extract was purified by a simple clean-up tube (MWCNTS-COOH, HLB, SG; weight ratio: 1:7.5:7.5). Then, the sample solution was filtered through the organic membrane (0.22 μm) for further determination on an Ultimate 3000 UPLC system (Thermo Fisher Scientific, San Jose, CA, USA) coupled to a TSQ Vantage TM triple-stage quadrupole mass spectrometer (Thermo Fisher Scientific, San Jose, CA, USA). Analytes were separated on an Agilent Extend C_18_ column (150 × 3.0 mm, 3.5 µm, Art. No. 763954-302) for gradient elution with methanol and water containing 5 mM ammonium acetate (0.35 mL·min^−1^, 30 °C, 10 μL) in terms of the following elution program: 15% B (initial), 15% B (0–1 min), 15–25% B (1–3 min), 25–50% B (3–6.5 min), 50–100% B (6.5–12 min), 100% B (12–15 min), 100–15% B (15–17 min), and 15% B (17–20 min).

### 4.3. Tea Consumption Data

The differences in tea samples, drinking habits, and preferences for various tea categories could influence the assessment of population exposure to mycotoxins through drinking tea. We divided the tea samples into four tea categories, i.e., green tea (23.01%), oolong tea (18.18%), black tea (29.55%), and dark tea (29.26%). To evaluate the risk of mycotoxin exposure when drinking different tea categories, we used tea consumption and body weight data from questionnaires obtained from previous studies (Appendix A) [27,41,42,43]. Specifically, we referred to the China Kadoorie Biobank (CKB) study which included a cohort of 512,891 adults from 5 urban areas and 5 rural areas in China, thus providing a complete database to support further research. Guan et al. focused on tea drinking behavior, and 512,824 samples were analyzed after the elimination of invalid data based on CKB. These collected data showed that tea drinkers in most of the project areas mainly drank green tea. Another questionnaire survey concerned the tea consumption of 219 residents from Kunming and Pu’er. Meanwhile, the relevant tea consumption data from Ulan Bator in Mongolia were also gathered by Ye et al. Participants in Kunming, Pu’er, and Ulan Bator in Mongolia mainly drank dark tea. In the latest national statistics for 2015–2019, more than 50% of total adults were overweight (BMI: 24.0–27.9) or obese (BMI > 28), with an average body weight of 64.3 kg [44]. The recommended dosage of green tea powder (1000 mg/day) was obtained from the USDA Dietary Supplement Ingredient Database (DSID; https://dsid.usda.nih.gov (accessed on 18 May 2022)).

### 4.4. Assessment of Health Risk

The risk of exposure to ingested mycotoxins by drinking green, oolong, black, and dark tea was assessed by point evaluation combined with the Monte Carlo assessment model.

#### 4.4.1. Health Risk: Deterministic Estimation

Tea consumption data and contamination levels in tea were used to assess the dietary risk of exposure to mycotoxins through tea drinking. The hazard quotient (HQ) was used to assess the noncarcinogenic risk of exposure to each mycotoxin (e.g., DON, ZEN, T-2, OTA, NEO, and CIT). The estimated daily intake (EDI, µg·kg^−1^ bw·day^−1^) of each mycotoxin and HQ were calculated using the following formulas:EDI = (C × CA)/BW(1)
HQ = EDI/PMTDI(2)
where C (µg·kg^−1^) is the average level of mycotoxins. In our study, if the pollutant level was not detected or lower than LOD, alternative values of 0, 1/2 LOD and LOD were applied to assess exposure. CA and BW represent daily tea consumption amount (g·person^−1^·day^−1^) and the body weight of participants (kg). The CA and BW values used in the model refer to previous studies. For Equation (2), the HQ is evaluated by the ratio of PMTDI (provisional maximal tolerable daily intake), which is proposed by the WHO (World Health Organization) and recommended by the USEPA (United States Environmental Protection Agency), and comes from the PTWI (provisional tolerable weekly intake). The PMTDI values of DON and its acetylated derivatives, ZEN and its modified forms, T-2, and OTA were 1.0, 0.25, 0.1, and 0.0143 µg·kg^−1^ bw·day^−1^, respectively [45,46,47]. The PTDI value of CIT was 0.2 µg·kg^−1^ bw·day^−1^ [48,49], and the PMTDI value of NEO referenced T-2. If HQ > 1, it means that mycotoxin is of concern. Otherwise, no significant health risk is found, meaning humans are not threatened by that exposure.

#### 4.4.2. Health Risk: Probabilistic Estimation

The Monte Carlo model is one of the applicable models to quantify and decrease the uncertainty in health risk assessments. In our study, we performed Monte Carlo simulations for probabilistic estimation with @-Risk Industrial 7.5 (Palisade Corporation, New York, NY, USA) software and Microsoft Excel 2016. The data on the amount of daily tea consumption, the mycotoxin content of the tea samples, and body weight were input into the @RISK software. The best distribution was chosen based on Kolmogorov–Smirnov and Anderson–Darling tests in the @Risk software. Monte Carlo simulations were performed using 10,000 iterations (confidence interval > 90%). The MOS (margin of safety) values at the mean and the 50th and 95th percentile (mean, P50, P95) were calculated using the following formula. P50 and P95, more statistically significant representatives in the distribution of exposure levels, were selected as middle and universal estimates, respectively:MOSi = Pi/PMTDI(3)

#### 4.4.3. Risk Ranking

Weight ratios of exposure to six groups of mycotoxins in four types of tea for the total population were regarded as the same in this work. The HQ values of mycotoxins from deterministic estimation were ranked. Meanwhile, the risk ranking could also be obtained by the MOS values of mycotoxins from the probabilistic evaluation. The HI (hazard index) value represents the combined noncarcinogenic risks of multiple mycotoxins. However, the interaction mechanisms and relevant coefficients among the six groups of mycotoxins were unknown and unavailable. In our study, the HI was calculated by adding the HQ or MOS for each group of mycotoxins from the deterministic and probabilistic estimation:HI= HQ_OTA_ + HQ_ZEN_ + HQ_DON_ + HQ_T-2_ + HQ_NEO_ + HQ_CIT_(4)
HI= MOS_OTA_ + MOS_ZEN_ + MOS_DON_ + MOS_T-2_ + MOS_NEO_ + MOS_CIT_(5)

Conventionally, an HI less than 1.0 indicates that the total exposure does not exceed the level considered “acceptable”, and people are unlikely to be exposed at a toxic level with possible consequences for health. In contrast, if it exceeds one, there is a possibility of suffering adverse effects.

#### 4.4.4. Risk Assessment for OTA and AFs

Considering the differences in potential nonneoplastic and neoplastic risk for OTA, the margin of exposure (MOE) using the benchmark dose (BMD) approach of the European Food Safety Authority was also applied. MOE was calculated using the following equations:MOE_1_ = nonneoplastic_BMDL10/ADD(6)
MOE_2_ = neoplastic_BMDL10/ADD(7)

BMDL10 values of 4.73 and 14.5 µg·kg^−1^ bw·day^−1^ were obtained from kidney lesions (nonneoplastic) observed in pigs and kidney tumors (neoplastic) seen in rats [50]. The ADD (average daily intake, mg·kg^−1^·day^−1^) data were from deterministic and probabilistic evaluations on OTA. In addition, MOE_1_ and MOE_2_ values exceeding 200 and 10,000 indicated no health concern.

AFs are considered to be carcinogens without associated PMTDI values to date. Thus, deterministic and probabilistic estimations could not be implemented for their risk assessment. In our study, the carcinogenic risk of individual lifetime AF exposure through tea consumption was estimated using the following formula [27,41]:R = ADD × SF(8)
where R is cancer rate for the individual lifetime, SF represents the carcinogenic slope factor (HBsAg^−^, 9 mg·kg^−1^·day^−1^), and ADD is the daily aflatoxin intake from deterministic (EDI) and probabilistic evaluation exposure (Pi) [27]. The acceptable level of carcinogenic risk for environmental risk management may change with the environmental policy. Considering the carcinogenic grade of AFs to humans, the lifetime cancer risk of AFs in this study was 10^−5^, which means one additional cancer in 100,000 humans ingesting tea containing aflatoxins over 70 years [41].

### 4.5. Data Analysis

Raw data from UPLC–MS/MS were recognized using Thermo Xcalibur Qual Browser 4.0., and GraphPad Prism 8.4 software was used for all statistical analysis and graphic production. A probability level of <0.01 was considered statistically significant.

## Figures and Tables

**Figure 1 toxins-14-00452-f001:**
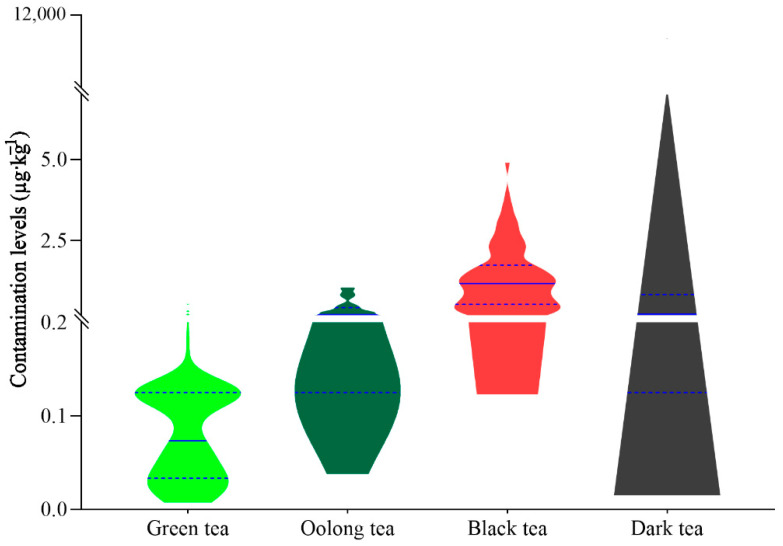
The distribution of contamination levels for OTA in different tea categories. The solid and dotted lines in the violin graphics represent the median and quartile distributions.

**Figure 2 toxins-14-00452-f002:**
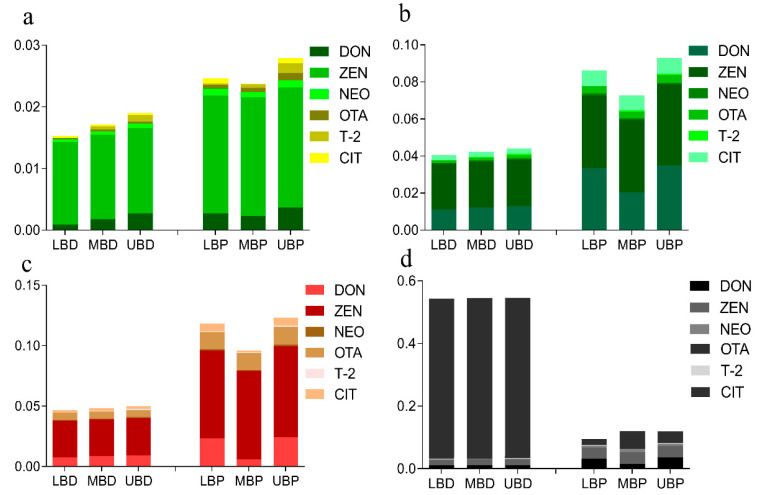
The (HI) values of six noncarcinogenic mycotoxins from tea consumption are the sum of the HQ values and MOS values (P95) from deterministic and probabilistic estimation with three bounds (lower, middle, and upper): (**a**) green tea, (**b**) oolong tea, (**c**) black tea, (**d**) dark tea. LBD/P: HQ/MOS values of deterministic/probabilistic estimation; MBD/P: HQ/MOS values of deterministic/probabilistic estimation; UBD/P: HQ/MOS values of deterministic/probabilistic estimation.

**Table 1 toxins-14-00452-t001:** The deterministic estimation of six groups of mycotoxin exposure for four types of tea consumption (µg·kg^−1^·day^−1^).

Mycotoxins	DONs	ZENs	NEO	OTA	T-2	CIT
PMTDI	1	0.25	0.1	0.0143	0.1	0.2
Lower Bound						
Green Tea	9.21 × 10^−4^	3.36 × 10^−3^	3.81 × 10^−5^	2.99 × 10^−6^	5.71 × 10^−6^	6.01 × 10^−5^
Oolong Tea	1.13 × 10^−2^	6.21 × 10^−3^	3.55 × 10^−5^	1.86 × 10^−5^	1.49 × 10^−6^	5.60 × 10^−4^
Black Tea	7.75 × 10^−3^	7.59 × 10^−3^	3.65 × 10^−5^	8.30 × 10^−5^	9.66 × 10^−6^	4.34 × 10^−4^
Dark Tea	1.07 × 10^−2^	4.42 × 10^−3^	2.50 × 10^−4^	7.23 × 10^−3^	6.35 × 10^−5^	1.16 × 10^−3^
Middle Bound						
Green Tea	1.81 × 10^−3^	3.41 × 10^−3^	5.31 × 10^−5^	4.53 × 10^−6^	5.29 × 10^−5^	6.20 × 10^−5^
Oolong Tea	1.21 × 10^−2^	6.26 × 10^−3^	5.50 × 10^−5^	1.97 × 10^−5^	4.78 × 10^−5^	5.61 × 10^−4^
Black Tea	8.55 × 10^−3^	7.64 × 10^−3^	5.84 × 10^−5^	8.31 × 10^−5^	5.62 × 10^−5^	4.35 × 10^−4^
Dark Tea	1.13 × 10^−2^	4.46 × 10^−3^	2.64 × 10^−4^	7.24 × 10^−3^	1.03 × 10^−4^	1.16 × 10^−3^
Upper Bound						
Green Tea	2.71 × 10^−3^	3.47 × 10^−3^	6.80 × 10^−5^	6.07 × 10^−6^	1.00 × 10^−4^	6.38 × 10^−5^
Oolong Tea	1.29 × 10^−2^	6.31 × 10^−3^	7.45 × 10^−5^	2.08 × 10^−5^	9.41 × 10^−5^	5.62 × 10^−4^
Black Tea	9.35 × 10^−3^	7.68 × 10^−3^	8.03 × 10^−5^	8.33 × 10^−5^	1.03 × 10^−4^	4.37 × 10^−4^

**Table 2 toxins-14-00452-t002:** The estimation of OTA exposure on four types of tea consumption with different methods based on exposure data from the upper bound deterministic and probabilistic estimation.

Tea	HQ/MOS	MOE_1_	MOE_2_
The upper bound deterministic estimation			
Green Tea	4.24 × 10^−4^	779,689.09	2,390,167.40
Oolong Tea	1.45 × 10^−3^	227,572.33	697,631.89
Black Tea	5.83 × 10^−3^	56,762.80	174,008.60
Dark Tea	0.51	653.59	2003.59
The upper bound probabilistic estimation (Mean)			
Green Tea	4.24 × 10^−4^	779,653.11	2,390,057.09
Oolong Tea	1.45 × 10^−3^	227,583.84	697,667.16
Black Tea	5.90 × 10^−3^	56,082.27	171,922.38
Dark Tea	8.12 × 10^−3^	40,746.62	124,910.34
The upper bound probabilistic estimation (P50)			
Green Tea	3.21 × 10^−4^	1,030,801.16	3,159,961.29
Oolong Tea	1.01 × 10^−3^	327,078.29	1,002,671.28
Black Tea	4.74 × 10^−3^	69,726.95	213,750.68
Dark Tea	1.21 × 10^−3^	273,674.19	838,958.92
The upper bound probabilistic estimation (P95)			
Green Tea	1.14 × 10^−3^	290,790.79	891,430.54
Oolong Tea	4.11 × 10^−3^	80,440.80	246,594.41
Black Tea	1.47 × 10^−2^	22,503.27	68,984.65
Dark Tea	1.90 × 10^−2^	17,411.17	53,374.64

**Table 3 toxins-14-00452-t003:** The upper bound deterministic and probabilistic estimation of carcinogenic risk (R) of aflatoxins from four types of tea consumption.

Mycotoxins	Deterministic Estimation	Probabilistic Estimation		
		Mean	P50	P95
AFB_1_				
Green Tea	2.05 × 10^−8^	1.86 × 10^−8^	1.86 × 10^−8^	2.29 × 10^−8^
Oolong Tea	1.69 × 10^−8^	1.69 × 10^−8^	1.69 × 10^−8^	1.69 × 10^−8^
Black Tea	1.75 × 10^−8^	1.75 × 10^−8^	1.73 × 10^−8^	1.86 × 10^−8^
Dark Tea	3.06 × 10^−8^	3.05 × 10^−8^	2.57 × 10^−8^	6.16 × 10^−8^
AFB_2_				
Green Tea	1.37 × 10^−8^	1.36 × 10^−8^	1.23 × 10^−8^	2.24 × 10^−8^
Oolong Tea	1.67 × 10^−8^	1.66 × 10^−8^	1.40 × 10^−8^	3.30 × 10^−8^
Black Tea	1.18 × 10^−8^	1.17 × 10^−8^	1.07 × 10^−8^	1.83 × 10^−8^
Dark Tea	2.77 × 10^−8^	2.75 × 10^−8^	2.13 × 10^−8^	6.78 × 10^−8^
AFG_1_				
Green Tea	1.80 × 10^−8^	1.80 × 10^−8^	1.75 × 10^−8^	3.01 × 10^−8^
Oolong Tea	1.55 × 10^−7^	1.56 × 10^−7^	3.59 × 10^−8^	6.75 × 10^−7^
Black Tea	3.29 × 10^−7^	3.26 × 10^−7^	2.30 × 10^−7^	9.49 × 10^−7^
Dark Tea	2.68 × 10^−7^	2.65 × 10^−7^	1.88 × 10^−7^	7.70 × 10^−7^
AFG_2_				
Green Tea	7.50 × 10^−7^	7.41 × 10^−7^	5.16 × 10^−7^	2.20 × 10^−6^
Oolong Tea	3.28 × 10^−6^	3.27 × 10^−6^	2.98 × 10^−6^	7.15 × 10^−6^
Black Tea	3.01 × 10^−6^	2.99 × 10^−6^	2.50 × 10^−6^	6.59 × 10^−6^
Dark Tea	4.99 × 10^−6^	5.07 × 10^−6^	4.64 × 10^−6^	**1.11 × 10^−5^**

The bold font represents R > 1.0 × 10^−5^.

**Table 4 toxins-14-00452-t004:** The upper bound probabilistic estimation of six groups of mycotoxin exposure on four types of tea consumption (µg·kg^−1^·day^−1^).

Mycotoxins	DONs	ZENs	NEO	OTA	T-2	CIT
PMTDI	1	0.25	0.1	0.0143	0.1	0.2
Mean						
Green Tea	2.31 × 10^−3^	3.46 × 10^−3^	5.81 × 10^−5^	6.07 × 10^−6^	1.05 × 10^−4^	6.31 × 10^−5^
Oolong Tea	1.28 × 10^−2^	6.49 × 10^−3^	5.06 × 10^−5^	2.08 × 10^−5^	9.41 × 10^−5^	5.53 × 10^−4^
Black Tea	9.26 × 10^−3^	7.68 × 10^−3^	8.00 × 10^−5^	8.43 × 10^−5^	9.40 × 10^−5^	4.33 × 10^−4^
Dark Tea	1.17 × 10^−2^	4.75 × 10^−3^	3.08 × 10^−4^	1.16 × 10^−4^	1.43 × 10^−4^	1.15 × 10^−3^
P50						
Green Tea	2.09 × 10^−3^	3.34 × 10^−3^	5.05 × 10^−5^	4.59 × 10^−6^	1.00 × 10^−4^	4.48 × 10^−5^
Oolong Tea	9.38 × 10^−3^	5.91 × 10^−3^	4.81 × 10^−5^	1.45 × 10^−5^	9.41 × 10^−5^	3.82 × 10^−4^
Black Tea	6.97 × 10^−3^	5.91 × 10^−3^	6.98 × 10^−5^	6.78 × 10^−5^	9.40 × 10^−5^	2.99 × 10^−4^
Dark Tea	6.60 × 10^−3^	4.02 × 10^−3^	6.79 × 10^−5^	1.73 × 10^−5^	1.24 × 10^−4^	7.96 × 10^−4^
P95						
Green Tea	3.68 × 10^−3^	4.88 × 10^−3^	1.16 × 10^−4^	1.63 × 10^−5^	1.59 × 10^−4^	1.82 × 10^−4^
Oolong Tea	3.49 × 10^−2^	1.10 × 10^−2^	7.29 × 10^−5^	5.89 × 10^−5^	9.43 × 10^−5^	1.67 × 10^−3^
Black Tea	2.42 × 10^−2^	1.88 × 10^−2^	1.46 × 10^−4^	2.10 × 10^−4^	1.14 × 10^−4^	1.30 × 10^−3^
Dark Tea	3.55 × 10^−2^	9.33 × 10^−3^	7.56 × 10^−4^	2.72 × 10^−4^	2.61 × 10^−4^	3.47 × 10^−3^

## Data Availability

The data that support the findings of this study are available from the corresponding author upon reasonable request.

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
