# Peer review of "Mycotoxins in Tea ((Camellia sinensis (L.) Kuntze)): Contamination and Dietary Exposure Profiling in the Chinese Population"

_toxins, 2022, doi:10.3390/toxins14070452_

Round 1

Reviewer 1 Report

L260 : As the results and conclusions are based on this sample selection, it might be good to add a better description of the used process. Was there a specific randomisation methodology used to select samples ? How do the authors assess the representativity of the selected 352 samples? How many does that represent compared to the total of available samples teas ? What about the Shangai Difute International Tea Compagny and the Yunnan Fengqing Logrun ? Are they representative tea sellers in China, in the world ?

L243 : It's a safe recommendation given by the authors to say that a better description of tea consumption might be need in the future to further assess health risk associated to tea consumption. It would also maybe be good to add a comment or recommendation to tea producers, based on the contamination level differences shown between green tea and black tea, to keep taking care of the used processess in order to keep the contamination levels as low as possible.

What the authors do think about adding a specific part of the survey in the future on the difference between organic and non organic teas (which is something that might influence the mycotoxin contamination levels) ?

Author Response

Dear Reviewer,

Thanks for your comments and suggestions to our manuscript (toxins-1765816). All the issues raised have been carefully addressed and all the changes have been clearly marked in the new revision. Here below are our descriptions for revising according to the point-to-point comments.

Yours sincerely,

Reviewer 2 Report

Dear Authors,

I believe the manuscript can rightly be considered for publication in Toxins.

There are only a few changes to be made:

TITLE _ Insert ((Camellia sinensis (L.) Kuntze)) instead of (Camellia sinensis L.)

LINE 114-117_ The HQ values reported in the text for green, oolong, black, and dark tea are not consistent with those in Table 1

The values in green tea decrease in the order ZEN>DON>T-2>NEO>CIT>OTA

The  values in oolong tea decrease in the order DON>ZEN>CIT>T-2>NEO>OTA

The  values in black tea decrease in the order DON>ZEN>CIT>T-2>OTA>NEO

The  values in dark tea decrease in the order DON>OTA>ZEN>CIT>NEO>T-2

I hope that my contribution may have been useful.

Best regards

Author Response

(The authors gave the same response as above.)

Reviewer 3 Report

The manuscript presents the mycotoxins contamination levels of different Chinese tea samples and uses these data for deterministic and probabilistic exposure assessment. The paper investigates many different aspects of exposure assessment giving a complete estimation of the possible risk associated to mycotoxins exposure associated to tea consumption. In addition, the paper is well written and clearly structured and limitations are highlighted. However, it should be mentioned in the results and discussion section, that tea is only one of the possible mycotoxins source in the diet, the overall exposure being the sum of different contributions from different sources.

Line 259 - Although the analytical procedure is described, a reference should be given for the method applied.

Author Response

(The authors gave the same response as above.)

Reviewer 4 Report

Dear authors, 

First of all, I would like to congratulate you for the work and suggest some modification in order to be accepted.

Comments:

-Correct all references number, they are superscript like "1"  but the authors' guide suggests that the reference must be cited like this: [1].

- Line 38 : Delete ")".

-Correct reference in line 90

-Line 96: (p≤ 0.01) insteed (P < 0.01). Correct it throughout paper. 

-Please rewrite the senteces in lines 98-99.

-Line 99 : " we did not detected high concentrations of ZEN..." the expression "high concentrations" is subgetive, so that I suggest authors to indicate the concentrations and better compare  both works.

- In general, I suggest reading the results and discussion section and improving the discussion. 

-Rewrite the senteces in lines 101-104.

-Figure 2: Include a dot.

-Correct table 3 title.

- Figure 1. : Include a dot after number and describe the figure. What does the Y-axis mean?

-Exclude lines 241-244 and include a section on conclusions, it is also recommended to formulate an exhaustive conclusion that responds to the proposed objectives.

-Include a quote in section 3.2.

Author Response

(The authors gave the same response as above.)
